High LYRM4-AS1 predicts poor prognosis in patients with glioma and correlates with immune infiltration

Wang Hai yue 1 2
Xie Ying 1 2
Du Hongzhen 1 2
Luo Bin 1 2 luobinyisheng@163.com
Li Zengning 1 2 lizengning@126.com
1 Department of Nutrition, The First Hospital of Hebei Medical University , Shijiazhuang, Hebei , China
2 Hebei Key Laboratory of Nutrition and Health , Shijiazhuang, Hebei , China
Uversky Vladimir
Electronic publication date: 2023 Oct 3
Publication date: 2023
Volume: 11
Electronic Location ID: e16104
Received 2023 Apr 21; Accepted 2023 Aug 25
Copyright: © 2023 Wang et al.
Copyright year: 2023
Copyright holder: Wang et al.
License: This is an open access article distributed under the terms of the Creative Commons Attribution License, which permits unrestricted use, distribution, reproduction and adaptation in any medium and for any purpose provided that it is properly attributed. For attribution, the original author(s), title, publication source (PeerJ) and either DOI or URL of the article must be cited.
License URL: https://creativecommons.org/licenses/by/4.0/

Keywords: LYRM4-AS1, Glioma, Prognosis, Immune environment, Immune infiltration

Funding: Research Projects of China Association for Geriatric Health Care HZ202102 Medical Science Research Projects in Hebei Province 20190454 “Spark” Youth Research Project of the First Hospital of Hebei Medical University XH201810 This work was supported by the Research Projects of China Association for Geriatric Health Care (No. HZ202102), the Medical Science Research Projects in Hebei Province (No. 20190454) and the “Spark” Youth Research Project of the First Hospital of Hebei Medical University (No. XH201810). The funders had no role in study design, data collection and analysis, decision to publish, or preparation of the manuscript.

==============================
Background

Many researches proved that non-coding RNAs are important in glioma development. We screened the differentially expressed genes through The Cancer Genome Atlas (TCGA) database and identified the molecule LYRM4-AS1 associated with prognosis. As a lncRNA, the expression level and role of LYRM4-AS1 in glioma are inconclusive. Therefore, we attempted to assess the clinical significance, expression and related mechanisms of LYRM4-AS1 in glioma by employing cell experiments and an integrative in silico methodology.

Methods

RNA-seq data were obtained from UCSC XENA and TCGA datasets. The Gene Expression Omnibus (GEO) database was used to download glioma-related expression profile data. The LYRM4-AS1 expression level was evaluated. Survival curves were constructed by the Kaplan–Meier method. Cox regression analysis was used to analyze independent variables. Patients were divided into high and low expression group base on the median LYRM4-AS1 expression value in glioma tissues. The DESeq2 R package was used to identify differentially expressed genes (DEGs) between two different expression LYRM4-AS1 groups. Gene set enrichment analysis (GSEA) was conducted. Next, the single-sample Gene Set Enrichment Analysis (ssGSEA) was done to quantify the immune infiltration of immune cells in glioma tissues. Gene expression profiles for glioma tumor tissues were used to quantify the relative enrichment score for each immune cell. Spearman correlation analysis was used to analyze the correlation between LYRM4-AS1 and biomarkers of immune cells as well as immune checkpoints in glioma. Finally, assays for cell apoptosis, cell viability and wound healing were conducted to evaluate the function on U87 MG and U251 cells after knocking down LYRM4-AS1.

Results

We found that LYRM4-AS1 was upregulated and related to the grade and malignancy of glioma. Survival analyses showed that high expression LYRM4-AS1 patients had poor clinical outcomes (P < 0.01). Cox regression analyses demonstrated that LYRM4-AS1 was an independent risk factor for overall survival (OS) in glioma (HR: 274 1.836; CI [1.278–2.639]; P = 0.001). Enrichment and immune infiltration analysis showed interferon signaling and cytokine-cytokine receptor interaction enriched in the LYRM4-AS1 high-expression phenotype, and LYRM4-AS1 showed significantly positively related to immune infiltration as well as immune checkpoints (P < 0.01). The knockdown of LYRM4-AS1 in U87 MG and U251 cells can inhibit migration and proliferation of cells (P < 0.05).

Conclusions

These findings indicated that the increased LYRM4-AS1 may be useful for the diagnosis and prognosis of glioma and might participate in the immune infiltration.

Introduction

The average annual age-adjusted incidence rate of all malignant and non-malignant brain and other central nervous system tumors was 24.71 per 100,000 population between 2015 and 2019 in the United States, in which approximately 28.3% were malignant and 71.7% were non-malignant (Ostrom et al., 2022). About 49% of malignant brain tumors were glioblastomas, and 30% were diffusely infiltrating lower-grade gliomas (Schaff & Mellinghoff, 2023). The overall survival of adult-type diffuse gliomas varies depending on the specific subtype and histological grade. Patients with IDH-mutated CNS WHO grade 2 astrocytomas survived significantly longer with a median survival up to 10 years, reduced to 5 years for grade 3 and even lower for grade 4 tumors (Antonelli & Poliani, 2022). Conventional treatment strategies for most glioblastomas, including surgery, radiation and chemotherapy, have showed limited efficacy. Recently, significant efforts have been reported to identify functional therapeutic targets for glioblastomas (Jin, Mao & Qiu, 2017; Li et al., 2023), new molecular targets and treatment techniques are still urgently needed to guide treatment and improve patient prognosis for glioblastomas.

Long noncoding RNA (lncRNA) has a length greater than 200 nucleotides. In tumors, RNA-seq has identified a number of lncRNAs that are abnormally expressed (Batista & Chang, 2013). Several researches have proved that in many biological processes, lncRNAs play a crucial role (Jariwala & Sarkar, 2016), including regulating tumorigenesis and tumor development. As a lncRNA, DILC can inhibit STAT-3 activating IL-6 signaling and prohibit liver cancer stem cell expansion (Wang et al., 2016). LINC00152 modulated glioblastoma (GBM) malignant progression and proneural-mesenchymal transition through the miR-612 dependent AKT2/NF-κB pathway (Cai et al., 2018). The lncRNA X inactive specific transcript (XIST) XIST can amplify the chemoresistance of glioma cell lines to temozolomide (TMZ) through directly targetting miR-29c via SP1 and MGMT (Du et al., 2017). Moreover, lncRNA-based therapy had achieved significant effects in GBM model mice (Kim et al., 2018), which suggested that lncRNA had a great potential therapeutic value. Therefore, the related mechanisms of lncRNA should therefore be investigated in order to provide references for glioma diagnosis. We screened differentially expressed lncRNAs through The Cancer Genome Atlas (TCGA) database and identified molecule LYRM4-AS1, which was thought to be a significant high-risk lncRNA in survival for glioma patients (Ouyang et al., 2022). LYRM4-AS1 is a lncRNA located on chromosome 6p25.1. At present, its expression level and biological function in glioma are unclear. Therefore, by analyzing bioinformatics data and conducting cell experiments, we explored whether LYRM4-AS1 is related to worse outcomes, and the involved mechanisms of LYRM4-AS1 in glioma had also been investigated.

In the current study, we found that high expressed LYRM4-AS1 was related to poorer survival and LYRM4-AS1 can be considered as an emerging prognostic biomarker of glioma. Gene set enrichment analysis (GSEA) showed high expression LYRM4-AS1 connected with P53 signaling pathway, signaling by NOTCH, cell cycle, JAK_STAT signaling pathway, and single-sample Gene Set Enrichment Analysis (ssGSEA) analysis and Spearman correlation analysis also suggested high LYRM4-AS1 expression level was correlated with immune infiltration, as well as the immune checkpoints in glioma. Finally, we knocked down LYRM4-AS1 in U87 MG and U251 glioma cells and found that low expressed LYRM4-AS1 in glioma cells can suppress the cell activity and migration.

Materials and Methods

A graphical illustration of the study is shown in Fig. 1.

Figure 1 Research flow of this study.

UCSC Xena, California Santa Cruz (UCSC) XENA dataset (https://xenabrowser.net/datapages/); TCGA, The Cancer Genome Atlas database (https://tcga-data.nci.nih.gov/); GEO, Gene Expression Omnibus database (https://www.ncbi.nlm.nih.gov/geo/); DEGs, Differentially expressed genes; GO, Gene Ontology function enrichment; KEGG, Kyoto Encyclopedia of Genes and Genomes pathway enrichment analysis; GSEA, Gene set enrichment analysis.

Datasets

The RNA-seq data through the Toil process of TCGA and Genotype-Tissue Expression (GTEx) processed into transcripts per million reads (TPM) format were collected from University of California Santa Cruz (UCSC) XENA dataset (Vivian et al., 2017). The RNA-seq data in HTSeq-FPKM and HTSeq-counts format with clinical information from GBM and brain lower grade glioma (LGG) projects were acquired from TCGA (https://tcga-data.nci.nih.gov/). The TPM data generated from level 3 HTSeq-FPKM data was used to analyze the expression of LYRM4-AS1 between tumor and normal tissues using Wilcoxon rank sum test. Unavailable or unknow clinical features in glioma tissues were regarded as missing values and the study met the publication guidelines stated by TCGA. P < 0.05 was considered statistically significant.

In addition, GEO database was used to download glioma-related expression profile data in the data format MINiML. Based on organism (homo sapiens), and experiment type (expression profiling), design (no additional processing was performed on the sample and there was a control group), and finally GSE15209, GSE16011 and GSE21354 were selected. GSE15209 (Pollard et al., 2009) was based on the GPL570 platform, and the expression data of 11 human fetal neural stem cell (NS) cell lines and nine human glioma neural stem cell (GNS) cell lines were selected from the dataset; GSE16011 (Gravendeel et al., 2009) was based on the GPL8542 platform, and the expression data of eight normal tissues and 276 glioma tissues were selected from the dataset; GSE21354 (Liu et al., 2011) was based on the GPL570 platform, and the expression data of three normal tissues and 14 glioma tissues were selected from the dataset.

R2 is a web-based genomics analysis, and visualization application platform (http://r2.amc.nl) developed by the Academic Medical Center in Amsterdam. R2 was used to investigate the relationship of LYRM4-AS1 expression and overall survival probability, and a Kaplan–Meier curve was plotted.

Differentially expressed genes (DEGs) analysis

The RNA-seq data collected from GBM and LGG projects of TCGA were used for expression research. The median LYRM4-AS1 expression value in glioma tissues was used as a cut-off value to divide patients into high and low expression groups. The RNA-seq data in HTSeq-counts format were compared between high and low LYRM4-AS1 expression groups to identify DEGs using the DESeq2 (Love, Huber & Anders, 2014) R package, and the thresholds were set at |log2-fold change (FC)| > 2.0 and adjusted P-value < 0.05.

GO and KEGG enrichment analysis

DEGs were used to conducted Gene Ontology (GO) function enrichment and Kyoto Encyclopedia of Genes and Genomes (KEGG) pathway enrichment analysis by the clusterProfiler (Yu et al., 2012) R package. Adjusted P-value < 0.05 was considered significant.

Gene set enrichment analysis

To detect the significant pathway and function difference in high and low LYRM4-AS1 expression groups, GSEA (Subramanian et al., 2005) was performed by clusterProfiler (Yu et al., 2012) R package. A function or pathway was considered significant enrichment with adjusted P-value < 0.05, false discovery rate (FDR) q-value < 0.25, and normalized enrichment score (|NES|) > 1.

Immune infiltration analysis

The ssGSEA was done by GSVA R package to quantify the immune infiltration in glioma tissues (Hanzelmann, Castelo & Guinney, 2013). According to the signature genes of the 24 types of immunocytes (Bindea et al., 2013), the relative enrichment score of every immune cell was quantified from the gene expression profile for each tumor tissue derived from the TCGA-GBM and TCGA-LGG projects. Correlation between immune cells and LYRM4-AS1 was analyzed by Spearman correlation, and the infiltration of immune cells between high and low LYRM4-AS1 expression groups was compared by Wilcoxon rank sum test.

Cell culture

U87 MG and U251 cells were purchased from Procell Co., Ltd. (Wuhan, China). The cells were cultured in Dulbecco’s modified Eagle’s medium (DMEM; Gibco, Billings, MT, United States) containing 10% fetal bovine serum (FBS; Biological Industries, Beit HaEmek, Israel), 100 U/mL penicillin (Solarbio, Beijing, China), and 100 μg/mL streptomycin (Solarbio, Beijing, China), and cultured in an incubator with 5% CO2 at 37 °C. Cells were passaged when they reached approximately 90% confluency.

Cell transfection

Small interfering (si)-lncRNA LYRM4-AS1 and si-negative control (NC) were designed by GenePharma Co., Ltd. (Suzhou, China). Transfection reagent was purchased from Qiagen (301005; Hilden, Germany). U87 MG and U251 were seeded in a 6-well plate (5 × 105/well), and the cells were transfected with 0.6 μg/mL si-lncRNA LYRM4-AS1 or si-NC. Transfection was done according to the fast-forward protocol of the transfection reagent manufacturer. After transfection for 8 h, the medium was replaced with complete medium. After treatment for another 48 h, total RNA of the cells with different treatments was isolated, and the expression level of LYRM4-AS1 was determined using RT-qPCR to assess the transfection efficiency. As shown in Table 1, si-lncRNA LYRM4-AS1 or si-NC sequences can be found.

Table 1 The sequences of primers or siRNA.

Primer or siRNA		Sequence (5′–3′)	
LYRM4-AS1	F	CTGAGCACGGCCCGAAAAGC	
	R	GACCAGTCTGGGCAACACAGC	
β-actin	F	TCAGGTCATCACTATCGGCAAT	
	R	AAAGAAAGGGTGTAAAACGCA	
siNC	Sense	UUCUCCGAACGUGUCACGUTT	
	Antisense	ACGUGACACGUUCGGAGAATT	
siLYRM4-AS1	Sense	GGUAUCACUGACUUCCUAATT	
	Antisense	UUAGGAAGUCAGUGAUACCTT	
siLYRM4-AS2	Sense	GCUUUCCUACUGUGGCCUUTT	
	Antisense	AAGGCCACAGUAGGAAAGCTT	

Cell viability assays

Cell viability of the U87 MG and U251 cells after treated with si-lncRNA LYRM4-AS1 or si-NC was measured using the Cell Counting Kit-8 (CCK-8; Solarbio, Beijing, China) (Fu et al., 2019). The cells were treated with different treatments, and then added 10 μL of reaction reagent. After incubation for 1, 2, 3, 4, 5, 6, 7 days, a microplate reader was used to measure absorbance at 450 nm (Multiskan FC; Thermo Fisher Scientific, Waltham, MA, USA).

RT-qPCR

Total RNA was extracted according to the manufacturer’s instructions by the RNA extraction kit (Solarbio, Beijing, China). Total RNA was reverse transcribed into cDNA using a cDNA Reverse Transcription Kit (Vazyme, Nanjing, China). RT-qPCR reaction conditions: predenaturation at 95 °C for 5 min; 40 cycles at 95 °C for 10 s and 60 °C for 30 s; melt curve at 95 °C for 15 s, 60 °C for 60 s and 95 °C for 15 s. The β-actin was used as an internal reference. The relative mRNA expression of LYRM4-AS1 was calculated using the 2−ΔΔCt method (Livak & Schmittgen, 2001). The sequences of primers are listed in Table 1.

Morphological changes due to apoptosis

Characteristic morphological changes associated with apoptosis were assessed by fluorescence microscopy using Hoechst 33258 (Hoechst 33258 Stain solution; Solarbio, Beijing, China) (Chen et al., 2020). U87 MG cells were seeded in a 6-well plate (5 × 105/well). After 72 h transfection, fluorescence microscope imaging was conducted after washing cells with PBS and staining them with Hoechst 33258 (10 μg/mL) for 10 min at 37 °C.

Flow cytometry detecting cell apoptosis

The qualitative apoptosis of U87 MG cells transfected with si-lncRNA LYRM4-AS1 or si-NC were determined using PE Annexin V Apoptosis Detection Kit (559763; BD Biosciences, Franklin Lakes, NJ, USA) by flow cytometry performed according to the manufacturer’s instruction. Briefly, the cells were suspended in 100 µl 1× binding buffer. Subsequently, 5 µl of Annexin V-PE was added, and the mixture was placed on ice for 15 min in the dark prior to the addition of 400 µl 1× binding buffer and 5 µl 7-AAD. The cells were resuspended in 1 ml PBS and analyzed using a FC-500 type flow cytometer (Beckman Coulter, Inc., Brea, CA, USA). Early and late apoptotic cells were assessed. The apoptotic rate was measured using the EXPO 32 ADC v1.2 software (Beckman Coulter, Inc., Brea, CA, USA) (Tang et al., 2019).

Wound healing assay

The migration ability of U87 MG and U251 cells were determined using wound healing assay (Hu et al., 2018). The cells were seeded in a 6-well plate (5 × 105/well). After 48 h transfection, cells were scratched along the marking line using a 10 μl tip to the plate and the floating cells were washed with PBS. Cell migrations were perceived by inverted microscope and photographed. The wound area and box height were calculated by Image Pro Plus 6.0 software (Media Cybernetics, Inc., Rockville, MD, USA). The relative of migration percentage was calculated as follows: relative of migration percentage = (scraped area – residual area)/scraped area * 100%.

Statistical analysis

The expression comparison of LYRM4-AS1 between tumor tissues and normal tissues was analyzed with the Wilcoxon rank sum test. Wilcoxon rank sum test and Kruskal–Wallis test were conducted to assess the relationships between clinicopathologic characteristics and the LYRM4-AS1 expression. The Pearson χ2 test was taken to analyze the association between clinicopathological characteristics and LYRM4-AS1 high and low expression levels with a cut-off of the median LYRM4-AS1 expression value. The Kaplan–Meier method was used to construct survival curves. A Cox regression analysis was performed on the independent variables. The XIANTAO platform (www.xiantao.love) was used to plot receiver operating characteristic (ROC) analysis to analyze the effectiveness of LYRM4-AS1 expression to discriminate glioma from normal tissues. Glioma survival rates over 1, 3, and 5 years was predicted using a nomogram by the XIANTAO platform (www.xiantao.love). Furthermore, Spearman correlation analysis was used to analyze the correlation between LYRM4-AS1 and biomarkers of immune cells as well as immune checkpoints in glioma using RNA-seq data from the TCGA-GBM and TCGA-LGG projects. The one-way ANOVA was used to compare cell viability, relative of migration percentage and the expression of LYRM4-AS1 across groups. All statistical tests were two-sided and P < 0.05 was considered statistically different.

Results

LYRM4-AS1 was high expressed in glioma

RNA-seq data from TCGA and UCSC Xena were extracted. LYRM4-AS1 expression level was analyzed between tumor and normal tissues. As shown in Fig. 2A, LYRM4-AS1 was significantly upregulated in many tumors, such as GBM, colon cancer, esophageal cancer, and hepatocellular carcinoma. Next, the different expression of LYRM4-AS1 in glioma was verified. The scatter plot shown that the level of LYRM4-AS1 was dramatically increased in glioma tissues compared with normal glioma tissues (Figs. 2B and 2C). To validate the expression of LYRM4-AS1, the database of GEO was used. And we found that LYRM4-AS1 was highly expressed in gliomas in GSE15209, GSE16011 and GSE21354 (Figs. 2E–2G). Finally, we evaluated the resolving ability of LYRM4-AS1 by ROC curve analysis. The AUC value was 0.939 (95% confidence interval CI [0.930–0.949]) (Fig. 2D), suggesting that LYRM4-AS1 could effectively distinguish normal tissue from glioma tissue.

Figure 2 The expressions of LYRM4-AS1 in glioma tissues.

RNA-seq data from UCSC XENA (A) datasets were used to analyze the expression of LYRM4-AS1 in cholangiocarcinoma (CHOL), esophageal cancer (ESCA), colon cancer (COAD), multiforming glioma (GBM), head and neck squamous cell carcinoma (HNSC), renal chromophobe cell carcinoma (KICH), hepatocellular carcinoma (LIHC), prostate cancer (PRAD), gastric cancer (STAD), thyroid cancer (THCA), lung adenocarcinoma (LUAD), Pheochromocytoma and paraganglioma (PCPG), lung squamous cell carcinoma (LUSC). The expression level of LYRM4-AS1 was analyzed in glioma tissues and in normal tissues based on RNA-seq data from TCGA (B) and UCSC XENA (C) datasets. The ROC analysis of LYRM4-AS1 using the RNA-seq data from TCGA-GBM and TCGA-LGG projects by XIANTAO platform (www.xiantao.love) (D). The expression level of LYRM4-AS1 was analyzed in GSE15209 (E), GSE16011 (F) and GSE21354 (G). Comparisons between the two groups were made using the Wilcoxon rank sum test. ns, P ≥ 0.05; *P < 0.05; **P < 0.01; ***P < 0.001.

LYRM4-AS1 expression correlated with clinical characteristics

The RNA-seq data and clinical characteristics (Ceccarelli et al., 2016) including gender, race, age, WHO grade, IDH status, epidermal growth factor receptor (EGFR) status, 1p/19q codeletion, histological type and primary therapy outcome were collected from GBM and LGG projects of TCGA. As showed in Figs. 3 and S1, the LYRM4-AS1 expression was significantly different among different clinicopathological characteristics, including WHO grade (P < 0.001), IDH status (P < 0.001), 1p/19q codeletion (P < 0.001), primary therapy outcome (P < 0.001), EGFR status (P < 0.001) and age (P < 0.001). In different histological types, LYRM4-AS1 expression in glioblastoma showed significantly different in astrocytoma, oligodendroglioma and oligoastrocytoma (P < 0.001). No differences of LYRM4-AS1 expression were found in gender and race (P > 0.05). To further verify the significance of LYRM4-AS1 expression, a correlation analysis was performed between high and low LYRM4-AS1 expression groups among different clinicopathological characteristics. The consistent results were obtained and summarized in Table 2.

Figure 3 The correlation analysis of LYRM4-AS1 expression and clinical characteristics of glioma patients using RNA-seq data with clinical information from TCGA-GBM and TCGA-LGG projects.

The LYRM4-AS1 expression in different clinicopathological characteristics of patients with glioma, including WHO grade (A), IDH status (P < 0.01) (B), 1p/19q codeletion (P < 0.01) (C), primary therapy outcome (P < 0.01) (D), histological type (P < 0.01) (E), EGFR status (P < 0.01) (F), age (P < 0.01) (G) and race (P = 0.19) (H). SD, stable disease; PR, partial response; CR, complete response; PD, progressive disease.

Table 2 The association between LYRM4-AS1 expression and the clinical characteristics of glioma patients.

Characters	Level	Low expression of LYRM4-AS1	High expression of LYRM4-AS1	P	
WHO grade (%)	G2	162 (55.3%)	54 (16.9%)	<0.001	
	G3	116 (39.6%)	121 (37.8%)		
	G4	15 (5.1%)	145 (45.3%)		
IDH status (%)	Mut	292 (88.2%)	132 (40.0%)	<0.001	
	WT	39 (11.8%)	198 (60.0%)		
1p/19q codeletion (%)	Codel	114 (34.1%)	54 (16.4%)	<0.001	
	Non-codel	220 (65.9%)	276 (83.6%)		
Primary therapy outcome (%)	CR	99 (35.5%)	36 (21.8%)	<0.001	
	PD	46 (16.5%)	57 (34.5%)		
	PR	46 (16.5%)	16 (9.7%)		
	SD	88 (31.5%)	56 (33.9%)		
Gender (%)	Female	146 (43.6%)	138 (41.2%)	0.584	
	Male	189 (56.4%)	197 (58.8%)		
Race (%)	Asian	4 (1.2%)	9 (2.7%)	0.133	
	Black or African American	12 (3.7%)	20 (6.0%)		
	White	311 (95.1%)	302 (91.2%)		
Histological type (%)	Astrocytoma	106 (31.6%)	86 (25.7%)	<0.001	
	Glioblastoma	15 (4.5%)	145 (43.3%)		
	Oligoastrocytoma	86 (25.7%)	42 (12.5%)		
	Oligodendroglioma	128 (38.2%)	62 (18.5%)		
EGFR status (%)	Mut	10 (3.0%)	63 (19.2%)	<0.001	
	WT	318 (97.0%)	265 (80.8%)		
Age (mean (SD))		42.32 (13.70)	51.39 (15.23)	<0.001	

High LYRM4-AS1 expression in glioma patients predicted worse prognosis

The Kaplan–Meier curves of overall survival (OS), disease-specific survival (DSS) and progression-free interval (PFI) in glioma patients were plotted to clarify the prognostic value of LYRM4-AS1 expression. According to Figs. 4A–4C, the high expression LYRM4-AS1 group showed worse OS (P < 0.001), DSS (P < 0.001) and PFI (P < 0.001) than the low expression LYRM4-AS1 group. The same analyses were conducted in different subgroups including age, WHO grade, IDH status, EGFR status, 1p/19q codeletion, primary therapy outcome and histological type. The results also showed LYRM4-AS1 high expression was also correlated with poor outcomes in different subgroups of glioma patients (Fig. S2). Finally, we analyzed the relationship between LYRM4-AS1 and patient prognosis using published data from glioma patients (Gravendeel et al., 2009) in the R2 database. Kaplan–Meier curve analysis found that patients with high expression of LYRM4-AS1 had a worse prognosis Fig. 4D.

Figure 4 Prognostic analysis of high LYRM4-AS1 expression in glioma patients.

Prognostic analysis of high LYRM4-AS1 expression in overall survival (A), progression-free interval (B) and disease-specific survival (C) of glioma patients based on survival data and RNA-seq data obtained from the TCGA-GBM and TCGA-LGG projects. The numbers below the figures denoted the number of patients at risk in each group. The prognostic analysis of high LYRM4-AS1 expression in overall survival (D) of glioma patients based on R2 database.

A high LYRM4-AS1 level was independent prognostic factor of OS in patients with glioma

Univariate and multivariate Cox regression analyses were conducted to explore whether LYRM4-AS1 was an independent prognostic factor for OS. As showed in Table 3, a univariate Cox regression analysis revealed high LYRM4-AS1 was connected with a short OS (hazard radio [HR]: 3.993; 95% CI [3.009–5.299]; P < 0.001), as well as WHO grade (HR: 0.170; CI [0.116–0.249]; P < 0.001), EGFR status (HR: 0.276; CI [0.203–0.374]; P < 0.001), age (HR: 0.212; CI [0.162–0.277]; P < 0.001), 1p/19q codeletion (HR: 0.216; CI [0.138–0.338]; P < 0.001), IDH status (HR: 9.850; CI [7.428–13.061]; P < 0.001). Furthermore, variables with P < 0.1 in the univariate Cox regression analysis including WHO grade, EGFR status, age, 1p/19q codeletion, IDH status and LYRM4-AS1 were included into multivariate Cox regression analysis, the result indicated that WHO grade (HR: 0.491; CI [0.314–0.766]; P = 0.002), age (HR: 0.517; CI [0.387–0.698]; P < 0.001), IDH status (HR: 5.009; CI [3.312–7.576]; P < 0.001) and LYRM4-AS1 (HR: 1.836; CI [1.278–2.639]; P = 0.001) could serve as independent factors for predicting OS of patients with glioma. To provide a quantitative method to predicting clinical outcome of glioma patients, these independent prognostic factors were incorporated to establish a nomogram, including WHO grade, IDH status, age and LYRM4-AS1. As shown in Fig. S3, the C-index of nomograms was 0.843 (95% CI [0.832–0.854]) suggesting a relatively precise predictive performance for the OS predictive nomograms. The calibration curve also suggested favourable consistency in the prediction and the observation.

Table 3 Univariate and multivariate Cox regression analysis of OS in glioma patients.

Characteristics	HR (95% CI) univariate analysis	P value Univariate analysis	HR (95% CI) multivariate analysis	P value multivariate analysis	
WHO grade (G2 vs. G3 & G4)	0.170 [0.116–0.249]	<0.001	0.491 [0.314–0.766]	0.002	
EGFR status (WT vs. Mut)	0.276 [0.203–0.374]	<0.001	1.297 [0.917–1.835]	0.141	
Race (Asian & Black or African American vs. White)	1.240 [0.757–2.032]	0.393			
Age (<=60 vs. >60)	0.212 [0.162–0.277]	<0.001	0.517 [0.383–0.698]	<0.001	
Gender (Female vs. Male)	0.813 [0.631–1.047]	0.109			
1p/19q codeletion (codel vs. non-codel)	0.216 [0.138–0.338]	<0.001	0.657 [0.387–1.116]	0.120	
IDH status (WT vs. Mut)	9.850 [7.428–13.061]	<0.001	5.009 [3.312–7.576]	<0.001	
LYRM4-AS1 (High vs. Low)	3.993 [3.009–5.299]	<0.001	1.836 [1.278–2.639]	0.001	

Identifications of DEGs and functional enrichment analyses

To further explore the biological mechanism of LYRM4-AS1 in glioma patients, DEGs were identified and their functions and interactions were also investigated in high and low LYRM4-AS1 expression groups. The volcano plot showed the identified 917 DEGs including 788 upregulated and 129 downregulated (|logFC| > 2 and adjust P < 0.05) (Fig. 5A and Table S1) and the heat-map showed the top 20 DEGs according to the |logFC| value containing 10 upregulated and 10 downregulated (Fig. 5B). Then, GO and KEGG functional enrichment analyses revealed the biological processes and molecular functions of LYRM4-AS1 significantly related to extracellular matrix organization, antimicrobial humoral response, receptor ligand activity, cytokine activity (Fig. 5C and Table S2), and cytokine-cytokine receptor inter-action, protein digestion and absorption, ECM-receptor interaction, IL-17 signaling pathway (Fig. 5D and Table S3). In order to further explore the biologic functions of LYRM4-AS1 in glioma, GSEA analysis was performed between two different expression LYRM4-AS1 groups using the curated gene set collection from MSigDB (c2.cp.v7.0.symbols). There were 236 pathways that showed significant enrichment in the LYRM4-AS1 high expression group (Table S4), including P53 signaling pathway, signaling by NOTCH, cell cycle, JAK_STAT signaling pathway, interferon signaling and cytokine-cytokine receptor interaction (Fig. S4). It can be indicated that the potential role of LYRM4-AS1 in the progression of glioma by regulating immune and inflammatory response, cell cycle and some important signal pathways.

Figure 5 Identification of DEGs and functional enrichment analyses.

Volcano plot showed the identified DEGs based on RNA-seq data collected from GBM and LGG projects of TCGA. Blue dots represented up-regulated DEGs, and red dots represented down-upregulated DEGs (A). Heat map showed the top 20 DEGs according to the |logFC| value containing 10 upregulated and 10 downregulated (B). The significantly enriched GO terms (C) and KEGG terms (D) are colored by adjusted P-value.

The correlation between LYRM4-AS1 and immune infiltration

To investigate the correlation between LYRM4-AS1 expression and immune infiltration, the LYRM4-AS1 expression level and immune cell enrichment were analyzed by ssGSEA method. As shown in Fig. 6, Th2 cells, macrophages, aDCs (activated DCs) and neutrophils were positively correlated with LYRM4-AS1 expression significantly (P < 0.001 of all), and the Spearman r were 0.460, 0.505, 0.401 and 0.613 respectively (Figs. 6B–6E). Wilcoxon rank sum test also showed infiltration levels of Th2 cells, macrophages, aDCs and neutrophils were significantly higher in the high expression LYRM4-AS1 groups when compared with low expression group (Fig. S5, P < 0.001 of all).

Figure 6 The correlation between LYRM4-AS1 expression and immune infiltration.

Correlation between the relative abundances of immune cells and LYRM4-AS1 expression level (A). The size of dots showed the absolute value of Spearman r. The correlations of aDCs (B), macrophages (C), neutrophils (D), Th2 cells (E) and LYRM4-AS1 expression. The markers of 24 types of immune cells were obtained from Bindea et al. (2013), and RNA-seq data of glioma patients were from GBM and LGG projects of TCGA.

The expression correlation of LYRM4-AS1 with biomarkers of immune cells will further explore the role of LYRM4-AS1 in tumor immunity. As showed in Table 4, a significant positive correlation was observed between LYRM4-AS1 and B cell’s biomarkers (CD19 and CD79A), CD4+ T cell’s biomarkers (CD4), M1 macrophage’s biomarkers (NOS2, IRF5, and PTGS2), M2 macrophage’s biomarkers (CD163, VSIG4, and MS4A4A), neutrophil’s biomarkers (CEACAM8, ITGAM and CCR7), T cell’s biomarkers (CD2, CD3E), Th1 cell’s biomarkers (TBX21, STAT4 and TNF), Th2 cell’s biomarkers (GATA3, STAT6, STAT5A and IL13) and Th17 cell’s biomarkers (STAT3 and IL17A) in glioma. The results of these studies partially support the claim that LYRM4-AS1 was positively linked to immune cell infiltration.

Table 4 Correlation analysis between LYRM4-AS1 and biomarkers of immune cells in GBM determined by TCGA database.

Immune cell	Biomarker	R value	P value	
B cell	CD19	0.31	<0.001	
CD79A	0.19	<0.001	
CD4+ T cell	CD4	0.41	<0.001	
M1 macrophage	NOS2	0.2	<0.001	
IRF5	0.34	<0.001	
PTGS2	0.21	<0.001	
M2 macrophage	CD163	0.46	<0.001	
VSIG4	0.37	<0.001	
MS4A4A	0.41	<0.001	
Neutrophil	CEACAM8	0.1	0.008	
ITGAM	0.33	<0.001	
CCR7	0.45	<0.001	
T cell	CD2	0.540	<0.001	
CD3E	0.5	<0.001	
Th1 cell	TBX21	0.46	<0.001	
STAT4	0.008	<0.001	
TNF	−0.019	0.624	
Th2 cell	GATA3	0.52	<0.001	
STAT6	0.29	<0.001	
STAT5A	0.44	<0.001	
IL13	−0.051	0.182	
Th17 cell	STAT3	0.65	<0.001	
IL17A	0.022	0.562	

The relationship between LYRM4-AS1 and immune checkpoints

To further explore the association of LYRM4-AS1 with immunity, the correlation analyses were explored between LYRM4-AS1 and immune checkpoints in glioma determined by TCGA database. The major immune checkpoints include PDCD-1, CD274 (also known as PD-L1), CTLA-4, HAVCR2 (also called TIM-3), LAG-3 and TIGIT which are responsible for tumor immune escape. As suggested in Fig. 7 and Table S5, the expression of LYRM4-AS1 was significantly correlated with these six major immune checkpoints.

Figure 7 The correlation between LYRM4-AS1 expression and immune checkpoints.

Spearman correlation of LYRM4-AS1 with expression of immune checkpoints in glioma (A). The expression correlation of LYRM4-AS1 with PDCD1 (B), CD274 (C), CTLA4 (D), LAG3 (E), TIGIT (F), and HAVCR2 (G). The RNA-seq data of glioma patients were from GBM and LGG projects of TCGA. ***P < 0.001.

Inhibition of LYRM4-AS1 expression can inhibit glioma cell viability and migration

Considering the correlation of LYRM4-AS1 with gliomas, we next examined the effect of LYRM4-AS1 on glioma cell viability. First, by transfecting siRNA into U87 MG and U251 cells, LYRM4-AS1 expression was found to be downregulated (Figs. 8C, 8D). Next, we detected the cell viability 1–7 days after knockdown of the LYRM4-AS1 and found that the cell viability decreased significantly on the 3rd day and continued to decrease in the following days (Figs. 8A, 8B). To investigate whether LYRM4-AS1 is involved in cell apoptosis, Hoechst was used to examine the morphologic changes in U87 MG cells after transfected with siRNA. As shown in Fig. 8G, bright staining and condensed nuclei were observed decreased when knockdown the LYRM4-AS1. Furthermore, we analyzed the apoptosis by flow cytometry. As shown in Fig. 8H, apoptotic cells were increased when knockdown the LYRM4-AS1 which indicated that LYRM4-AS1 could promote cell apoptosis in U87 MG. Then, the migration ability of U87 MG and U251 were investigated. As indicated by the wound healing assay, LYRM4-AS1 knockdown inhibited U87 MG and U251 cell migration ability compared with control (Figs. 8E, 8F). These results suggested that knockdown LYRM4-AS1can inhibit glioma cell viability and migration.

Figure 8 The role of LYRM4-AS1 expression on glioma cell viability and migration.

The cell viability of U87 MG and U251 cells after transfected with siLYRM4-AS1 or siNC was determined by CCK8 assays at indicated time points (A and B). RT qPCR analysis of LYRM4-AS1 in U251 (C) and U87 MG (D) cells after 48 h of transfection with siLYRM4-AS1 or siNC. Expression was normalized to cells transduced with the siNC in cells. Images and quantitative results of wound healing assay of U251 (E) and U87 MG (F) cells after 48 h of transfection. Hoechst 33258 nucleus staining (G) and Annexin V stain by flow cytometry (H) in U87 MG cells after 48 h of transfection with siLYRM4-AS1 or siNC. Data presented are means ± SD from three independent experiments. *P < 0.05.

Discussion

Due to the heterogeneity and complex pathogenesis of glioma, the 5-year survival rate has not much improved although the development of new therapies has greatly ameliorated the conditions of patients with glioma. Thus, it is essential to find new diagnostic markers and therapeutic targets for glioma. In recent years, researches related to lncRNA have attracted the attention of many cancer fields. The aberrant lncRNA expression may contribute to tumorigenesis and some lncRNAs have become tumor-specific markers. Some studies showed that the different expression of lncRNA involved in tumorigenesis, tumor development and pathological processes of glioma (Han et al., 2012), and cancer patients’ lncRNA may be able to predict their prognosis (Li et al., 2014). Another study found that lncRNA MEG3 was downregulated in glioma and contributed to regulation of cancer cell apoptosis and proliferation (Wang, Ren & Sun, 2012). These findings indicated that lncRNA may be a potential prognostic factor and therapeutic target in patients with glioma, which offers new ideas for effective treatment of glioma patients.

LYRM4-AS1 is a lncRNA located on chromosome 6p25.1, and is an antisense RNA of LYRM4. A newly discovered gene for mitochondrial diseases, LYRM4, encodes ISD11 protein, which acts as an iron-sulfur cluster (Lim et al., 2013). However, the studies of the role of LYRM4-AS1 have rarely been reported. In this study, LYRM4-AS1 expression was analyzed and the results showed LYRM4-AS1 upregulated in glioma (Fig. 2). Next, the clinical significance of the abnormal LYRM4-AS1 expression was explored. Isocitrate dehydrogenase (IDH) is a key player in the tricarboxylic acid (TCA) cycle and the mutation status of IDH has a significant impact on metabolism processes of glioma. IDH status can affect DNA methylation (Turkalp, Karamchandani & Das, 2014), which resulted in a worse prognosis. Studies also found that IDH mutations usually characterize LGG, which is weakly invasive and usually results in a better prognosis (Sun et al., 2013; Wang et al., 2020). Whereas, IDH wild-type LGGs exhibited a more invasive feature and a poor prognosis (Dunn, Andronesi & Cahill, 2013). The loci on distal arm of chromosome 1p frequently more occur loss in oligodendroglioma which is an important type of LGG (Qi et al., 2020) than in GBM (Di et al., 2018). Studies have confirmed that codeletion of 1p/19q enhanced the sensitivity of tumor cells to radiotherapy and chemotherapy, and was associated with favourable clinic outcomes (Kaloshi et al., 2008). EGFR encodes a transmembrane glycoprotein, which can bind to epidermal growth factor (EGF) and locate on the cell surface. The protein promotes tumor proliferation and invasion by activating signal transduction factor, such as RAS. Studies of glioma showed that EGFR was associated with the degree of malignancy and angiogenesis of tumor (Lal et al., 2002). In our study, LYRM4-AS1 showed high expression in subgroups of WHO grade 4, IDH wild status, 1p/19q non-codeletion (Fig. 3). These results indicated that LYRM4-AS1 may contribute to poor clinical outcome of glioma.

In order to analyzing the connection between LYRM4-AS1 expression level and clinical outcome of patients with glioma, a prognosis association analysis was performed. We found that LYRM4-AS1 was an independent risk factor for predicting the prognosis of patients with glioma (Table 2) and high LYRM4-AS1 expression had a worse prognosis than patients with low LYRM4-AS1 expression (Fig. 3). Our finding was consistent with the studies of Ouyang et al. (2022) and Cao et al. (2023), suggesting that LYRM4-AS1 was a high-risk lncRNA in survival for glioma patients.

Inflammatory cytokines exhibit a vital role in the development of cancer (Almeida et al., 2019). Previous studies have reported that inflammatory and immune responses contributed to the pathogenesis of glioma (Antonios et al., 2017). There are relatively few studies on the mechanism of role of LYRM4-AS1, and a recent report found that the beneficial effects of the bone marrow mesenchymal stem cell-derived exosomes on the pathogenesis of osteoarthritis through targeting the LYRM4-AS1/GRPR/miR-6515-5p signaling pathway. LYRM4-AS1 was highly expressed in inflammatory cell models (Wang et al., 2021). In our study, GO and KEGG analysis results showed the high LYRM4-AS1 expression group was significantly associated with the cytokine activity and cytokine-cytokine receptor interaction (Fig. 5), which was confirmed by the GSEA results (Fig. S4). These findings suggested that LYRM4-AS1 may play a role in regulating inflammation and immune responses during the development of glioma.

Another important aspect of this study was that LYRM4-AS1 was found to be correlated with immune infiltration. The tumor microenvironment composed of many types of immune cells is characterized as hypoxic and acidic, which induces inflammatory response and recruits immune cell infiltration (Franklin et al., 2014). Macrophage is a powerful modulator of immune responses in tumor microenvironments. It can promote tumor progression by suppressing immune system and associated with anti-chemotherapy drugs. According to the secreted cytokines, helper T cells can further differentiate into Type-1 helper T cells (Th1 cells) and Type-2 helper T cells (Th2 cells). Cytokines including IL-13, IL-5 and IL-4 are mainly secreted by Th2 cells, which can stimulate humoral immune response. Imbalance of the Th1/Th2 is often observed in cancer patients, which is connected with immune escape in tumor (Sasaki et al., 2009). Neutrophils are one of the most abundant leukocytes, which account for 70% of total leukocytes and are the first lines of defense (Shaul et al., 2016). Studies demonstrated that the patients of glioma with high level of neutrophils had a shorter survival time (Joseph, Sawant & Rajarathnam, 2017), suggesting neutrophils may correlated with immunosuppressed in tumor. Dendritic cells are the most powerful antigen-presenting cells (APC) in body and provide immune regulatory signals through cell-to-cell contact and cytokines (Macri et al., 2018; Merad et al., 2013). Mature DC cells can effectively activate initial T cells and are at the center of initiating, regulating, and maintaining immune responses. Under stimulation by inflammatory, DCs were activated to initiate and promoted immunity (Bharadwaj et al., 2010). In the field of tumor immunotherapy research, the six immune checkpoints PDCD1, CD274, CTLA-4, LAG-3, TIGIT, and HAVCR2 can inhibit the proliferation, activation and effector functions of T cells, and maintain the immune homeostasis in the body. In our study, there was a positive correlation between high LYRM4-AS1 expression and Th2 cells, macrophages, neutrophils, and aDCs (Figs. 6, S5), as well as the major six immune checkpoints (Fig. 7, Table S5). These findings further verified that LYRM4-AS1 may play an essential role in glioma and associate with immune infiltration. Finally, we experimentally verified that knockdown of LYRM4-AS1 can indeed inhibit cell activity and migration ability of U87 MG and U251 glioma cells.

Although this study enhances our knowledge of the role of LYRM4-AS1 in glioma, there are still some limitations. Although we did some cell experiments, analyses mainly based on public databases are not enough for understanding the molecular mechanisms of LYRM4-AS1 in glioma. In order to confirm LYRM4-AS1 expression and function, further cell and clinical studied should be conducted. Furthermore, retrospective studies have limits, especially in lacking some information and non-uniform intervening measures. So, more experiments and prospective studies are needed in the sooner future.

In conclusion, LYRM4-AS1 may be a potential prognostic molecular marker of poor survival in glioma, which might be associated with inflammation and immune infiltration This study provides new and promising insights for the diagnosis and treatment of glioma. Further researches are required to identify the underlying mechanism of LYRM4-AS1 to effectively evaluate patient’s survival and improve treatment of glioma.

Conclusions

In summary, we elucidate the potential of LYRM4-AS1 in prognostic prediction in glioma patients and the mechanisms involved may involve immune infiltration. Our current findings provide a new potential target for the treatment of glioma.

Supplemental Information

Supplemental Information 1 The correlation analysis of LYRM4-AS1 expression and gender of glioma patients.

The RNA-seq data with clinical information were obtained from TCGA-GBM and TCGA-LGG projects, and correlation of LYRM4-AS1 expression and gender of glioma patients was analyzed.

Click here for additional data file.

Supplemental Information 2 Prognostic analysis of high LYRM4-AS1 expression in glioma patients.

Prognostic analysis of high LYRM4-AS1 expression in IDH status:Mut (A), WHO grade: G3 (B), 1p/19q codeletion: Non-codel (C), Age: ≤60 (D, Age: > 60 (E), Histological type: Astrocytoma (F), Primary therapy outcome: PD (G), Primary therapy outcome: SD&CR&PR (H) of glioma patients. SD, stable disease; PR, partial response; CR, complete response; PD, progressive disease. The RNA-seq data with clinical information were obtained from TCGA-GBM and TCGA-LGG projects.

Click here for additional data file.

Supplemental Information 3 Construction of an OS predictive nomogram based on LYRM4-AS1 expression and clinicopathological factors.

The OS nomogram based on WHO grade, IDH status, age and LYRM4-AS1 was constructed (A). The C-index of the nomograms was 0.843 (95% CI [0.832–0.854]). Drawing a vertical line from the total point axis straight downward to the outcome axis can pick up the probability of patients with glioma at 1,3,5-years. The calibration curve nomogram predicting 5-year OS (B). The horizontal coordinate was the survival probability predicted by the OS nomogram and the vertical coordinate is the actual survival probability. The 5-year survival predicted was indicated by the blue line and the gray line represented the ideal situation in which predicted and actual survival coincide. The closer the predicted line was to the diagonal line, the more accurate the model was.

Click here for additional data file.

Supplemental Information 4 LYRM4-AS1 related signaling pathways based on GSEA.

Pathways and biological processes were differentially enriched in LYRM4-AS1-related phenotype, including P53 signaling pathway (A), signaling by NOTCH (B), cell cycle (C), JAK_STAT signaling pathway (D), interferon signaling (E) and cytokine-cytokine receptor interaction (F). The top portion showed the enrichment scores. If the normalized enrichment score (NES) was positive, a peak appeared on the left side, indicating that the core molecules of the gene set were mainly concentrated in the high expression group on the left side. Each vertical line in the middle represented one molecule in the gene set. The lower part visualized the values after normalizing the gene set expression data. The RNA-seq data were obtained from TCGA-GBM and TCGA-LGG projects.

Click here for additional data file.

Supplemental Information 5 The infiltration levels of Th2 cells, Macrophages, aDCs and Neutrophils in high and low LYRM4-AS1 expression group.

The analysis of LYRM4-AS1 expression levels and immune cell infiltration levels of Th2 cells (A), Macrophages (B), aDCs (C) and Neutrophils (D) compared with low LYRM4-AS1 expression groups. RNA-seq data of glioma patients were from GBM and LGG projects of TCGA.

Click here for additional data file.

Supplemental Information 6 The identified DEGs between high and low LYRM4-AS1 expression group of glioma tissues.

Click here for additional data file.

Supplemental Information 7 The results of the GO enrichment analysis.

Click here for additional data file.

Supplemental Information 8 The results of the KEGG enrichment analysis.

Click here for additional data file.

Supplemental Information 10 The results of GSEA analysis.

Click here for additional data file.

Supplemental Information 11 Correlation of LYRM4-AS1 expression with immune checkpoints.

Click here for additional data file.

Supplemental Information 12 R code.

Click here for additional data file.

Supplemental Information 13 Raw data: cck-8.

Click here for additional data file.

Supplemental Information 14 Raw data: qPCR.

Click here for additional data file.

Supplemental Information 15 Raw data: wound healing assay.

Click here for additional data file.

Supplemental Information 16 Raw data: wound healing pictures.

Click here for additional data file.

Abbreviations

CCK-8 Cell Counting Kit-8

DEGs Differentially Expressed Genes

DSS Disease-Specific Survival

EGF Epidermal Growth Factor

EGFR Epidermal Growth Factor Receptor

FC Fold Change

FDR False Discovery Rate

GBM Glioblastoma

GEO Gene Expression Omnibus

GO Gene Ontology

GSEA Gene Set Enrichment Analysis

GTEx Genotype-Tissue Expression

HR Hazard Radio

IDH Isocitrate Dehydrogenase

KEGG Kyoto Encyclopedia of Genes and Genomes

LGG Brain Lower Grade Glioma

NES Normalized Enrichment Score

OS Overall Survival

PFI Progression-Free Interval

ROC Receiver Operating Characteristic

ssGSEA single-sample Gene Set Enrichment Analysis

TCA Tricarboxylic Acid

TCGA The Cancer Genome Atlas

Th1 cells Type-1 helper T cells

Th2 cells Type-2 helper T cells

TPM Transcripts Per Million reads

UCSC XENA University of California Santa Cruz XENA dataset

Additional Information and Declarations

Competing Interests

Author Contributions

Data Availability

The authors declare that they have no competing interests.

Hai Yue Wang performed the experiments, analyzed the data, prepared figures and/or tables, and approved the final draft.

Ying Xie performed the experiments, prepared figures and/or tables, and approved the final draft.

Hongzhen Du analyzed the data, prepared figures and/or tables, and approved the final draft.

Bin Luo conceived and designed the experiments, authored or reviewed drafts of the article, and approved the final draft.

Zengning Li conceived and designed the experiments, authored or reviewed drafts of the article, and approved the final draft.

The following information was supplied regarding data availability:

The raw data and R code are available in the Supplemental Files.

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
