# Peer review of "High LYRM4-AS1 predicts poor prognosis in patients with glioma and correlates with immune infiltration"

_PeerJ, doi:10.7717/peerj.16104_

## Round 0.1 · original submission · Major Revisions

Please address the concerns of all reviewers and amend the manuscript accordingly.

Reviewer 1 ·

Basic reporting

1. It is better to use abbreviation list in MS.
3. Please insert key words.

Experimental design

4. Insert a reference for analyzing method in Real time PCR section.
5. It is better to insert a reference for any part of wound healing assay, Cell apoptosis assay, Cell viability assays.

Validity of the findings

2. Please insert p-value in where mentioned the results of abstract was significant.

Reviewer 2 ·

Basic reporting

In this article, the authors addressed relevant information about the status of the lncRNA LYRM4-AS1 on different types of gliomas. They present how it is expressed between the different WHO grades of gliomas and the possible biological processes in which LYRM4-AS1 is involved, emphasizing immune tumor evasion. They also perform functional assays in two human glioblastoma cell lines. Although this topic could be of interest in the research field, some issues need to be made before the acceptance of the manuscript:
1. It comes to my attention that the authors mentioned in lines 44 and 45: “Our previous research screened the differentially expressed genes through …” Also, in the introduction section, lines 97 to 100, the authors mentioned a previous work in which they found LYRM4-AS1 as an important prognostic marker from gliomas. However, they do not cite such previous work of them or other authors. Neither do they describe in this paper how the preliminary screening was made. The authors could improve the hypothesis of their study by showing or citing this.
2. Abstract section:
a. Why do the authors mention just TCGA when they use more than one database (like UCSC)?
3. Introduction:
a. Although the Introduction is now of sufficient extension, some changes must be made to improve the precision of the knowledge presented about gliomas. First, the epidemiological data could be improved with more recent references because the authors cite data from 2018. In addition, the authors place the statement that the mortality and disability caused by gliomas are high and that such tumors are highly invasive and aggressive. However, this is a partial truth. Just WHO 3 and 4-grade gliomas are such aggressive and invasive tumors, while patients diagnosed with WHO 1-2 usually have better prognostic and survival times. Also, the age and incidence rates between LGG and WHO 3 and 4 Gliomas are different. I suggest the authors review the CBTRUS reports and the manuscript of Antonelli M et al., 2022 about the new 2021 WHO classification. It comes to my attention that authors used some terms that the new 2021 WHO classification has now eliminated, like glioblastoma multiforme.
b. The authors cite how some lncRNAs are involved in regulating many biological processes. However, it would be enlightening if they briefly explain how lncRNA executes such regulation (their mechanisms of action).
c. It is unclear what the authors want to say with the statement: “After preliminary analysis, we found that LYRM4-AS1 showed a good prognostic significance.” In lines 105-106, the authors also indicate: “Our results indicated that high expressed LYRM4-AS1 was related to poorer survival”. Can you please clarify?
4. The phrase “LYRM4-AS1 corrected with P53 signaling pathway” is unclear. Can you please clarify?
5. The English language should be improved. Many punctuation issues in the manuscript need to be solved: It is uncommon to use apostrophes in research articles, like in line 78 of the manuscript. Besides, there are many sites where spaces have to be added before a parenthesis, like in lines 93, 94, 95, 131, and 145 among other lines. In the Abstract, the authors did not define the meaning of OS, and de word gliom seems to be misspelled. In the Introduction, the authors use the acronym GBM, but they did not define it before. In line 169, the number 2 of CO2 must be in a subscript font rather than a superscript. Please, capitalize the word “Wound” in line 200. In line 228: Did you mean glioblastoma multiforme instead of multiforme glioma? In line 375: did you mean Epidermal?

Experimental design

The original data presented in this manuscript are in line with the aims and scope of the journal, and the research question is well-defined and meaningful. Materials and methods are well described. However, some technical issues need to be addressed:
1. Materials and methods:
a. Authors must add the catalog number and/or the name of the si-lncRNA and transfection reagents. It is also not clear if the authors design their si-lncRNA.
b. It is unclear if, after 48h transfection when authors made the wound healing assay, the levels of LYRM4-AS1 are still low. Did authors check the levels of the LYRM4-AS1 at the time wound healing assays were performed?
c. It is unclear what were the criteria for choosing the GSE15209, GSE16011, and GSE21354 data sets.
2. It is not clear what are the characteristics of the called “normal tissues” chosen by the authors. Does it is normal tissue adjacent to the tumor? Does it is normal tissue from cadavers with specific characteristics?
3. It is unclear why the authors decided to evaluate the role of LYRM4-AS1 on viability and migration in vitro models when all their previous in silico analyses demonstrate the role of such lncRNA in controlling tumor immune escape. Could you please address this point? Is there another technique that authors could perform to demonstrate in vivo the role of LYRM4-AS1 in controlling tumor immune escape?

Validity of the findings

The authors presented sufficient material to evaluate their results. However, the authors must improve their results description. Also, the present discussion is a synthesis of the results that should be improved by discussing the possible mechanism (or mechanisms) by which LYRM4-AS1 regulates the immune tumor escape, cell viability, and migration.
1. Results:
a. In Figure 3, the authors did not indicate the meaning of the acronyms SD, PR, CR, and PD. Also, in the figure legend, a symbol must be removed after each parenthesis.
b. It is not clear how the percentages were calculated in Table 2. Also, it is not clear what is considered 100% by the authors. For example, in the character of 1p/19q, how does 83.6% of non-codel present a high expression of LYRM4-AS1, but 65.9% present a low expression?
c. In addition, not in the methods nor the results section explained what criteria the authors use for defining what is considered high or low levels of LYRM4-AS1.
d. It is not clear the relevance of the comparisons of LYRM4-AS1 vs IDH status, 1p/19q codeletion (Figure 3B and C; Supplementary Figure 2). Such characteristics, as Antonelli et al., 2022 described, are key characteristics that distinguish astrocytomas from glioblastomas, and oligodendrogliomas from other forms of gliomas, respectively, and authors had presented more specifically the status of LYRM4-AS1 expression with respect of the type of glioma (Figure 3E). Could the authors explain this?
e. Neither in the Results nor the figure legend the authors indicated which dataset was employed to generate the results in Figure 3.
f. Graphs B, C, and D of supplementary Figure 2 seem to be the same or very similar.
g. Regarding the differentially expressed genes (DEGS) analysis, many of the enriched processes in the condition of highly expressed LYRM4-AS1 are upregulated in WHO grade 3 and 4 gliomas, so it is not clear if such enrichment is just a correlation between LYRM4-AS1 and the processes in Figure 5D or if there is a causality process. How could the authors ensure that LYRM4-AS1 is affecting the expression of the identified 917 DEGs including 788 upregulated and 129 downregulated?
2. Discussion:
a. It is unclear what the authors are referring to in line 358, in which they said they explored the role of LYRM4-AS1 in glioma for the first time, and in line 382, they also said that their results were consistent with previous studies. Besides, despite the statement in line 382, the authors did not cite the studies they were referring to.
b. In line 383, the authors said that LYRM4-AS1 is related to the tumor grade, but it is unclear what kind of relation there is or with which type of glioma LYRM4-AS1 is more related.

Reviewer 3 ·

Basic reporting

The overall language is acceptable. The background context provided is satisfactory. Below are some comments addressing limitations in reporting.
- Please check the relevance of the references cited in the introduction and discussion. For example, in lines 82-84, "Despite their ability to prolong overall survival, these therapies often produce unsatisfactory therapeutic effects because of the antiapoptotic activity and serious systemic side effects of malignant tumors (Jiang et al. 2017; Wang et al. 2013)". Briefly looking into the cited articles, Jiang et al. 2017 and Wang et al. 2013, neither address the clinical treatment of glioma nor appear to support the claims made.
- The title's phrase '… correlates with immune response' may be ambiguous in interpretation and misleading, as the term 'immune response' encompasses a wide range of processes. The manuscript merely described a correlation with immune infiltration rather than a broader immune response. Please ensure the title accurately reflects the study's main findings.
- For all figure captions – avoid interpreting the data. The figure captions only provide more information to describe the figure, and interpretations should not be made. Statements like "LYRM4-AS1 significantly upregulated in […]" should not be in the figure caption. It was already described in the main text in the section beginning on line 225.
- In Figure 2B and 2D captions – from my understanding, UCSC Xena is an exploration tool for data and should not be cited as a data source. From the label in the figure, it appears the data came from a combination of TCGA and GTEx. It also appears from the axis labels that TCGA and GTEx data were used in normal tissue, and TCGA only was used for the tumor tissue. Please clarify the source of the data and adjust the figure captions accordingly.
- Figures 2A/2B and 2C/2D display the same data for tumor tissue samples but different data for normal tissue samples. Please explain the duplication of figures and clarify the differences between the datasets used for normal tissue samples in these figures.
- Figure 2A-B: most text labels are not legible. Please use higher-resolution images with larger texts. The statistical tests performed were also not described in the figure caption.
- Figure 2C-E: the texts are legible when zoomed in but are very small. Please enhance the readability of the text labels for the graphs.
- Figure 2E appears to be missing a figure caption. It seems to have been mislabeled as the caption for panel H. Also, it might help to provide additional details about the data source used in generating the ROC curves to ensure reproducibility.
- Line 237-249: it might help to include references specific to the panel labels in Figure 3. Also, include the source of the data in the Figure captions.
- Figure captions should be provided for supplemental figures. It will help with the interpretation of these figures.
- Figure 4D: the text annotations in the upper right corner of the figure were not visible.
- Line 281: "DGEs" spelling.
- For Figure 5, please indicate, in the figure caption (and preferably with more details in the Methods section) the source of the gene expression data and what comparisons were made.
- The text labels on Figures 5B and 5C are illegible due to blurriness, making it impossible to review these panels. Please improve the image quality.
- Figure 5C, which appears to display co-expression genes, is mentioned in the manuscript but lacks an appropriate interpretation in the Results section. Please verify this and either provide a suitable discussion for this figure panel in the manuscript or remove it if it is irrelevant to the study.
- Please provide higher-resolution images for Figure 6B-6D, as the text displaying the Spearman statistics is currently not legible.
- Figure 8 shows shRNA knockdown, but the figure caption, Results section (line 330), and the methods show siRNA. Please clarify if shRNA or siRNA knockdown was performed.
- Table 1: Forward (F) and Reverse (R) primer sequences for qRT-PCR are appropriately presented. However, the same F and R labels are shown for siRNA. Please clarify. The terms "Forward" and "Reverse" do not apply to siRNA. Please indicate the specific target sequence for the siRNA.
- The Discussion section should be made more concise. The current version is lengthy and contains numerous references to figures and tables to describe the results. The primary purpose of the Discussion section is to contextualize the findings and interpret their significance. Detailed descriptions of the specific results should be more appropriately placed in the Results section. Please revise the Discussion section to focus on interpreting and contextualizing the findings, and consider moving any detailed descriptions of the results to the appropriate sections in the manuscript.

Experimental design

Some significant limitations to the study design may need to be addressed.
- From the axis labels on Figure 2B and 2D, it is evident that TCGA and GTEx data are mixed in the normal groups, and TCGA-only data was used in the tumor groups. It is important to note that data from varying sources may have been generated from different sequencing platforms and processing pipelines, which may result in differences in data normalization. Also, because of the different sources, there may be batch effects. Please ensure that these potential limitations are addressed. Please also check the appropriate sections to make sure that the results from these data are not misinterpreted.
- All data sources need to be more clearly pointed out and cited. Particularly for Figures 5-7. This will help with the reproducibility of the analysis.
- Please provide a description of the basal levels of LYRM4-AS1 expression in the U251 and U87 MG cell lines used in the study. Additionally, it would also be helpful to explain if there is a particular reason for the selection of these specific model cell lines.
- The decrease in the expression of LYRM4-AS1 is not very significant with knockdown. Although it did result in observed phenotypes, the difference in expression in control vs. knockdown is not as significant as that reported in similar articles. It might help to optimize the transfection.
- The viability of cells with siRNA knockdown was measured for 7 days, which is close to the upper end of the duration of siRNA-mediated knockdown effects. However, according to the methods, the RT-qPCR for assessing the knockdown was performed at 48h (line 176). To ensure that the knockdown effects persist throughout the entire duration of the experiment, please consider performing an additional RT-qPCR at 7 days post-transfection (endpoint of the viability experiment) to validate the presence of the knockdown effects at that time point.
- Line 171: the section on cell transfection – cite a particular catalog number for Qiagen's transfection reagent and specify if the transfection was done per manufacturer protocol.
- LYRM4-AS1 overexpression experiments could be considered for further validation.

Validity of the findings

There are some significant limitations to the validity of some findings. Below are some comments.
- As previously discussed in the Experimental Design section, the combination of TCGA and GTEx data raises concerns due to potential differences in sequencing platforms, processing pipelines, and batch effects. Please be cautious in interpreting results derived from this mixed dataset and ensure that appropriate statistical methods are employed to account for any discrepancies that may arise from using data from different sources.
- The immune system checkpoints section (lines 320-327) is mostly driven by speculation based on a statistical relationship between the immune checkpoints. As correlation does not necessarily mean causation – the interpretation of the data needs to be careful. The claim that "There is evidence that LYRM4-AS1 may mediate glioma carcinogenesis through tumor immune escape" is not appropriate unless supported by experimental validation.
- The initial scratch at 0h appears to be significantly smaller in the "shNC" compared to the "shLYRM4-AS1" scratches in both panels of Figure 8F. This may significantly affect the results obtained. If possible, use specialized scratch assay apparatus instead of a pipette tip. Please provide data with similar initial scratch sizes.
- The raw wound healing assay (scratch assay) data provided in the PowerPoint file in the supplemental files are not appropriately labeled. To ensure accurate assessment, please provide raw images of the initial scratch at t=0h and the endpoint at t=24h, including 3 biological replicates, with proper labeling.
- Line 335: "As shown in Fig. 8 E, bright staining and condensed nuclei were observed decreased when knockdown the LYRM4-AS1, which indicated that LYRM4-AS1 could promote cell apoptosis in U87 MG" – the results in Figure 8E is not clear and is not sufficient to justify this claim. Additional experiments, such as observing the level of cleaved caspase-3, Annexin V stain, etc., are required to justify the observation of increased apoptosis.

Additional comments

A recent article published in April 2023 by Cao et al. [PMID: 37152037] appears to have also implicated LYRM4-AS1 in glioma, which may be of interest to the authors.

---

## Round 0.2 · Minor Revisions

Please address remaining issues pointed by the reviewer and revise manuscript accordingly.

Reviewer 3 ·

Basic reporting

The basic reporting of the manuscript has improved, and the authors have addressed most of the comments from the last round of review. However, there are still some concerns. Please see below for point-by-point comments.
- The rebuttal letter suggests that the source of the data in Figure 2B is the UCSC Toil RNAseq Recompute Compendium. If that is the case, the data source should be cited as the UCSC Toil RNAseq Recompute Compendium, not UCSC XENA (as described in the Figure caption of Figure 2B). UCSC XENA is a data visualization tool and may not be appropriately cited as a data source. Further, since the Toil data used is a meta-analysis that includes TCGA, there appears to be some redundancy between the TCGA data (Figure 2A) and Toil data (Figure 2B). The authors could consider removing Figure 2A to avoid any potential confusion.
- Figure 2E caption was included at the end of the Figure 2 captions. It should preferably be put in alphabetical order.
- Table 1: siRNA sense and antisense do not seem right. The sequences do not seem to be sense and antisense pairs of each other. There also appears to be a “V” in the antisense of siLYRM4-AS1 - which doesn't make sense for a nucleotide sequence. Please check and verify that the correct information is provided.
- For experiments using siRNA knockdowns in Figure 8, it might help to indicate, in the figure caption, the time post-transfection when the experiment was conducted.
- It may help to enlarge the Figure 3 text labels. Some texts are difficult to read.
- The STRING PPI data appear to have been removed in this revised version of the manuscript. However, it was still mentioned in Figure 1, the Methods and Abbreviations sections. Please check and revise as needed.

Experimental design

no comment

Validity of the findings

The authors addressed most of the concerns in the previous round of review. However, there are still a few points of concern.
- Line 346-347: “There was evidence that LYRM4-AS1 was linked to tumor immune escape.” – the conclusion of tumor immune escape may not be fully justified by the data presented. The data only suggest a correlation with immune infiltration. Please check and revise as necessary.
- The data presented only suggests a correlation with “immune infiltration” and does not directly imply “immune efficacy” (line 74, 418). Conclusions on immune efficacy based on immune infiltration data may be premature and not fully justified.

---

## Round 0.3 · accepted · Accept

All remaining concerns were adequately addressed and the manuscript was revised accordingly.